# Soy Protein Isolate as Emulsifier of Nanoemulsified Beverages: Rheological and Physical Evaluation

**DOI:** 10.3390/foods12030507

**Published:** 2023-01-22

**Authors:** Daniel Castro-Criado, Mercedes Jiménez-Rosado, Víctor Perez-Puyana, Alberto Romero

**Affiliations:** 1Department of Chemical Engineering, Escuela Politécnica Superior, 41011 Sevilla, Spain; 2Department of Chemical Engineering, Facultad de Química, 41012 Sevilla, Spain

**Keywords:** nanoemulsions, functional food, bioactive compounds, soy protein isolate

## Abstract

The production of biologically active molecules or the addition of new bioactive ingredients in foods, thereby producing functional foods, has been improved with nanoemulsion technology. In this sense, the aim of this work was to develop nanoemulsified beverages as potential candidates for the encapsulation of bioactive compounds, whose integrity and release across the intestinal tract are controlled by the structure and stability of the interfaces. To achieve this, firstly, a by-product rich-in protein has been evaluated as a potential candidate to act as an emulsifier (chemical content, amino acid composition, solubility, ζ-potential and surface tension were evaluated). Later, emulsions with different soy protein isolate concentrations (0.5, 1.0, 1.5 and 2.0 wt%), pH values (2, 4, 6 and 8) and homogenization pressures (100, 120 and 140 PSI) were prepared using a high-pressure homogenizer after a pre-emulsion formation. Physical (stability via Backscattering and drop size evolution) and rheological (including interfacial analysis) characterizations of emulsions were carried out to characterize their potential as delivery emulsion systems. According to the results obtained, the nanoemulsions showed the best stability when the protein concentration was 2.0 wt%, pH 2.0 and 120 PSI was applied as homogenization pressure.

## 1. Introduction

Nowadays, there is a new trend towards healthier diets that allow for better quality of life and longer life expectancy [1]. Consequently, the population consumes an increasing amount of healthier products, such as fruits and vegetables. However, another landmark that is highlighting this trend is the increased consumption of functional foods [2]. Functional foods are those that, in addition to their nutritional value, contain biologically active components that provide some additional and beneficial effect on health and reduce the risk of contracting certain diseases [3,4].

The food industry encounters problems when processing these functional foods, since most of them contain polyphenols, vitamins, carotenoids or probiotics that are highly degradable under environmental conditions, making them difficult to preserve [5,6]. Numerous studies have attempted to improve the conservation of these ingredients in order to improve the quality of functional food. For example, Pascuta et al. used polysaccharide-based edible gels to encapsulate them [7]. On the other hand, Tomasevic et al. used 3D printing as a novel tool to preserve the characteristics of these foods [8]. Nevertheless, the most interesting alternative is the use of nanoemulsions [9,10,11].

A nanoemulsion is a mixture of at least one immiscible liquid and a surfactant (nonionic or polymeric), typically in the form of tiny droplets between the size of 20 and 200 nm [12,13]. Nanoemulsions have several benefits over traditional emulsions. Firstly, they are optically transparent, making them potential candidates for inclusion in beverages that are clear. Secondly, nanoemulsions are colloidal systems that are kinetically stable. Thirdly, they have a large active surface area, which could increase their functionality [14]. In this way, they are able to encapsulate, protect and release functional compounds [15]. For the formation of nanoemulsions, external shear must be applied to break the larger droplets into smaller ones. Compared to microemulsions, very little is known about the creation and control of nanoemulsions, mainly due to the extreme shear that must be applied, which is beyond the reach of ordinary devices, to overcome surface tension effects and break the droplets sown to the nano-scale regime [9]. To this end, specialized equipment is often used, such as high-pressure homogenizers and ultrasounds capable of generating large disruptive mechanical forces that induce droplet breakage [14,16].

On the other hand, nanoemulsions need an emulsifier to maintain their stability for a long time. An emulsifier is a substance that helps in the mixing of two substances that are usually poorly miscible or difficult to mix, acting in the interphase [17]. Its nanostructure can also assist in regulating lipid digestion and the bioavailability of the bioactive compounds that are encapsulated by acting as a physical barrier that prevents the enzymes from getting close to the emulsified lipids. The interactions between emulsion droplets and, ultimately, the digestion of the interfacial film by gastrointestinal enzymes are influenced by the structure, composition and features of the interfacial layers, thus the study of the emulsifier’s surface and interfacial properties is needed to understand the behavior of the nanoemulsion [18]. Emulsifiers used in the food industry can be either synthetic, such as sorbitan esters and their ethoxylates, and sucrose esters [19], or natural, where the use of proteins stands out [20]. These have an amphiphilic character; that is, they have hydrophilic and hydrophobic groups that allow stabilizing the interface between aqueous and oily phases [21]. These emulsifiers are beneficial, since they enhance the natural character of these beverages, avoiding the use of synthetic substances that have been controversial in recent years for being a possible cause of allergies and diseases [22,23]. In addition, there are numerous food by-products that are rich in proteins, being a potential alternative for use as emulsifiers due to their low price. Among them, soy protein isolate is a potential alternative. This raw material comes from soy oil production as a by-product. To date, 60% of soybean production (353 million tons in 2020) is used for the production of soy oil [24]. This process generates 75% of soy protein isolate, thus it is a by-product that is produced in large quantities. However, its main use is for animal consumption, thereby being undervalued, thus its use as an emulsifier can generate added value [25].

In this way, the aim of this work was to design a nanoemulsified (oil-in-water) beverage using soy protein isolate as an emulsifier. The novelty of this project is the use of a soy by-product, which improves its added value. The soy by-product and its chemical (chemical and amino acid composition), physical (ζ-potential and solubility) and surface properties (interfacial tension and interfacial rheology) associated with its nutritional value, emulsifying capacity and stability were evaluated. The study in aqueous phase is important to verify the dispersion of the protein and whether it has a good capacity to emulsify and form the nanoemulsion (when the ζ-potential is zero, the protein is too aggregated, and, if it has low solubility, it does not facilitate its placement at the interface). Finally, the low interfacial tension allows the emulsifier to move to the interface, and the interfacial rheology shows the stability of the emulsion, favoring the resistance against destabilization phenomena (especially the breakage of the interface leading to coalescence). Later, nanoemulsion processing was optimized using different protein concentrations, pH values and homogenization pressures.

## 2. Materials and Methods

### 2.1. Materials

A soy protein isolate, supplied by Protein Technologies International (Supro 500E, Belgium) in the form of a yellowish powder with a grain size of 120–240 µm, was used as an emulsifier. The oil used for the development of the nanoemulsion was a refined sunflower seed oil obtained from a local market. Distilled and milli-Q water was used for the preparation of all solutions. During the development of the experiment, various reagents were used for pH modification and adjustment, such as HCl and NaOH. All of them were supplied by Sigma Aldrich (Saint Louis, MO, USA) in analytical grade.

### 2.2. Chemical Composition of Protein Isolate

The protein contents were determined in triplicate as %N × 6.25 using a LECO CHNS-932 nitrogen micro analyzer (Leco Corp., St. Joseph, MI, USA) [26]. Lipid content was analyzed by Soxhlet extraction [27]. Moisture and ashes were determined in triplicate by AOAC approved methods [28].

### 2.3. Amino Acid Composition

The amino acid composition of the protein isolate was determined using a method previously described by Felix et al. [29]. The sample was dissolved in 6 M HCl and incubated in an oven at 110 °C for 24 h. After the samples had been hydrolyzed, the pH was brought down to 7 using 6 M NaOH and they were then filtered through Whatman No. 1 glass microfiber filters. Finally, the sample was diluted with double distilled water at a ratio of 1:500. Subsequently, the material was examined using reversed-phase liquid chromatography using pre-column fluorescence derivation with o-phthaldialdeyde. A LC-9A liquid chromatograph with RF-530 fluoresce HPLC monitor and SIL-9A automated injector (Shimadzu Corp., Kyoto, Japan) was used for this purpose.

### 2.4. ζ-Potential

The isoelectric point of the protein in solution was determined by surface charge detection at different pH values using a Zetasizer Nano ZS zeta potential analyzer (Malvern Instruments, Malvern, UK) [30]. To this end, different samples were prepared by dispersing them in water at a concentration of 2.0 wt% at different pH values. This measurement provides insight into the stability of a particle and indicates the potential required to penetrate the surrounding ion layer of the particle to destabilize it. The point at which this occurs is called the isoelectric point, with this pH being the value at which the molecule dissociates equally, and solubility is practically null [31].

### 2.5. Solubility

To determine the solubility of the protein fraction, Markwell’s method [32] was followed, at different pH values (2–12). For this purpose, 1 g of the sample was dispersed in 40 mL of buffer at different pH values. The dispersions were kept in agitation for 30 min at 500 rpm to favor protein solubility. Subsequently, these were centrifuged for 20 min at 15,000 rpm at 10 °C. The supernatants were analyzed to determine the amount of protein dissolved at each pH value. For this, 3 mL of reagent C, which was prepared by mixing reagent A, consisting of 30 g of Na_2_CO_2_, 4 g of NaOH, 1.6 g of sodium-potassium tartrate and 10 g of SDS dissolved in 1 L of milli-Q water, and reagent B, consisting of 2 g of CuSO_4_ dissolved in 50 mL of distilled water, in a ratio of 100:1, was added to 1 mL of the sample. Subsequently, 0.3 mL of Folin-Ciocalteau’s reagent previously diluted in distilled water in 1:1 ratio was added. The sample was incubated for 45 min in the absence of light and stained blue due to the formation of a complex between the soluble proteins and the Folin-Ciocalteau’s reagent. This staining was measured using a Genesys-20 spectrophotometer (Thermo Scientific, Waltham, MA, USA) at a wavelength of 660 nm. For this evaluation, a calibration curve with bovine serum albumin was performed in advance.

### 2.6. Surface Tension

Surface tension behavior is similar to those obtained for the interfacial tension of the emulsions [33]. In this way, it was determined in order to evaluate the most suitable pH and protein concentration for the development of emulsions.

The surface tension was determined using the dynamic Wilhelmy platinum plate method [34]. For this, the plate is inserted into the sample and slowly raised until the plate is completely separated from the liquid. Before this occurs, the interface is bent on both sides of the plate. The stress exerted by the interface at this point is measured and it is balanced with the lifting force of the plate.

To perform the measurements, dispersions of the protein isolate were prepared with 20 mL of Milli-Q water at different concentrations (0.5 wt%, 1.0, wt%, 2.0 wt% and 4.0 wt%) and pH 2 as the zeta potential; as is described in the results section, in this case, these systems were very different from zero and were thereby expected to be more stable. Dispersions were also carried out at 2 wt% at different pH values (2, 4, 6 and 8). Measurements were carried out using a Sigma 701 tensiometer (KSV, Castlemead, UK).

### 2.7. Interfacial Rheology

The evaluation of the rheological properties was carried out to verify that the interfaces were structured, in order to avoid effects such as coalescence, which would destabilize the emulsion [35]. The rheological properties were measured at the interface of the solution at different concentrations (between 0.5 wt% and 2.0 wt%) at pH 2. This rheological study was carried out by means of dynamic oscillatory tests using a DHR3 rheometer (TA Instruments, New Castle, DE, USA) and a ring-shaped accessory [36]. For this, the previously prepared solution was deposited after a rest time (at least 30 min) and the ring was placed in the interface between the solution and the oil. The results were obtained by means of frequency sweep tests at room temperature (20 ± 2 °C) in an interval between 0.01 and 0.4 Hz in the linear viscoelastic interval, for which strain sweep tests were previously carried out at 0.1 Hz in order to determine the critical strain (the last strain in the linear viscoelastic interval).

### 2.8. Preparation of the Nanoemulsion

Emulsification is the operation by which two immiscible liquids are intimately mixed [37]. There are numerous methods for this purpose and, in general, it is achieved by applying high mechanical energy; in this case, vigorous agitation was used, which causes the droplets to deform and break into smaller and more finely dispersed droplets. The nanoemulsion was prepared in two phases: a first pre-emulsification phase and a second emulsification phase.

#### 2.8.1. Pre-Emulsion Preparation

To generate the emulsion, it was firstly pre-emulsified using a Rotor/Stator L5M mixer (Silverson, UK) [38]. The protein was dissolved in distilled water at different concentrations (0.5 wt%, 1.0 wt%, 1.5 wt% and 2.0 wt%) and kept in agitation for 1 h. After this time, the pH was measured and adjusted to values of 2, 3, 4, 5, 6, 7 and 8 depending on the sample using HCl 5 N or NaOH 1 M. Subsequently, an organic phase/aqueous phase ratio of 1:9 was selected, for which 100 g of aqueous phase and 11.11 g of organic phase were weighed. This is performed for two main reasons: on the one hand, the homogenizer can become clogged if the amount of oil in the emulsion is higher and, on the other hand, according to Yuan et al. (2008) [39], this ratio is very close to the optimum for this type of food emulsions. Once both phases have been prepared, the following steps are carried out to obtain the pre-emulsion: (1) the mixer speed is adjusted to 5000 rpm; (2) the aqueous phase is introduced into the coarse emulsion tube; (3) the base of the equipment is adjusted, and the agitation is activated; (4) the oily phase is added slowly while stirring for the correct emulsification of the phases for 1 min; and (5) once the oil is added, agitation is maintained for 5 min.

#### 2.8.2. Emulsion Preparation

The previously obtained pre-emulsion is taken to a piston homogenizer M-110P (Microfluidics, Westwood, MA, USA), in which the nanoemulsion is obtained [40]. For this, the pre-emulsion is introduced into the equipment by passing it through once at different pressures (80 PSI, 100 PSI, 120 PSI and 140 PSI) to reach the nanoemulsion with 50–200 nm droplets. The outlet temperature is measured. The nanoemulsion obtained is stored in an airtight flask in which sodium azide is added, to avoid the proliferation of microorganisms, and it is then kept refrigerated (4 ± 2 °C).

### 2.9. Nanoemulsion Characterization

#### 2.9.1. BackScattering

The stability of the obtained nanoemulsions was evaluated using a Turbiscan Classic MA2000 transmitted and scattered light analyzer (Formulaction, Toulouse, France). This analyzer allows quantifying destabilization by analyzing the light transmitted and scattered by the emulsion as a function of time [41,42]. For this analysis, cylindrical glass tubes of 15 cm in height and 16 mm in internal diameter were filled with 7 mL of the nanoemulsion. The optical reader scanned the total height of the sample, taking measurements of the transmitted and scattered intensity every 40 µm. The software then provided the transmittance and backscattering curves as a function of the sample height in millimetres. In addition, sweeps were carried out at certain time intervals in order to obtain the emulsions’ time evolution.

#### 2.9.2. Microstructure

The microstructure of the samples was determined by droplet size distribution analysis using a MasterSizer 2000 (Malvern Instruments, Malvern, UK) [43]. For this purpose, this equipment employs a laser diffraction technique by measuring the intensity of scattered light as the laser beam passes through the sample [44]. Measurements were made in duplicate using aliquots of the prepared nanoemulsions diluted in solutions of pH 8 with 0.05 M Tris-HCl and 1% SDS to promote flocculation. Measurements were made with 45 and 300 mm lenses during the 30 days after the preparation of the emulsions to observe the droplet size distribution as a function of time. From this analysis, data such as distribution uniformity (*U*), Equation (1), Sauter diameter (D_3,2_) and volume mean diameter (D_4,3_) were extracted.
(1)$$U=\frac{\sum V_{i}\left|d\left(v,0.5\right)-d_{i}\right|}{d\left(v,0.5\right)\sum V_{i}}$$
where *d*(*v*, 0.5) is the median of the distribution and *V_i_* is the volume of the drops with diameter *d_i_*.

#### 2.9.3. Rheology

To determine the rheological properties, a Haake MARS II rheometer (Haake, Vreden, Germany) was used. All measurements were carried out at 25 °C (controlled by a Peltier) using a 25 mm diameter plate-plate system with rough surfaces to avoid slippage problems and 500 µm of gap. Firstly, stress sweep tests were performed to determine the linear viscoelastic range and, subsequently, frequency sweep tests were carried out between 0.003 and 5 Hz. These measurements were carried out in duplicate at different times to observe their evolution. Thus, we obtained data that allowed the elastic (G′) and viscous (G″) moduli to be plotted against stress (τ) and frequency (ω).

### 2.10. Statistical Analysis

At least 3 replicates of each sample were performed in order to evaluate the replicability and reproducibility of the results. The results were expressed as mean value with its standard deviation. Significant differences were evaluated with *t*-tests at a confidence level of 95% (*p* < 0.05).

## 3. Results & Discussion

### 3.1. Chemical Composition of the Protein Isolate

Table 1 shows the results obtained after the chemical characterization of the raw material used. Most of the content (90.0 wt% on a dry basis) is associated with proteins, which allows classifying it as a protein isolate according to Pearson’s classification [45]. Proteins, as was mentioned above, are amphiphilic. In this way, they act as good emulsifiers by stabilizing the interface between the aqueous and lipid phases of an emulsion, creating a film between both phases [21]. Thus, this by-product could have a potential value to act as emulsifier in emulsions, as other by-products evaluated in the literature [46], such as potato protein isolate (89.1 wt% protein), which was used by Romero et al. [47], and pea protein isolate (93.0 wt% protein) [48]. This protein content has also been observed in other studies carried out with soy by-products, as is the case of Jiménez-Rosado et al. [49], where 91.0% protein was obtained. This by-product also has certain minority components that do not play a fundamental role in obtaining the desired emulsion.

Table 2 shows the amino acid composition of the soy protein isolate. As can be seen, the majority (57.8%) of the amino acids present are hydrophilic (they have a high affinity for water). This quality can favor protein solubility in the aqueous phase and its integration into the system. On the other hand, this protein also presents hydrophobic amino acids (28.8%), which grant the aforementioned amphiphilic behavior, which will help stabilize the emulsion. Among the amino acids, glutamic acid stands out. It may act as an anticarcinogen [50]. Another amino acid that should be mentioned is alanine, which increases the intramuscular carnosine content [51]. These two amino acids increase the added value of the use of this protein as an emulsifier in this beverage.

### 3.2. Characterization of the Aqueous Phase

#### 3.2.1. ζ-Potential and Solubility

The variation of both ζ-potential and solubility as a function of pH is shown in Figure 1.

Soy protein isolate presents good solubility, reaching 100% of solubility at high pH values (10–11.5), as was predicted due to their high content of hydrophilic amino acids. Nevertheless, soy protein solubility presents a typical U-shaped behavior between pH 2 and 10, with a minimum value between pH 4 and 5. This behavior is similar to that observed in previous studies carried out by other authors [52,53], and it is due to the charges present in the protein chains. Thus, the closer the pH value is to the isoelectric point (zero charge point), the fewer ionizable groups are present in the protein, making it more insoluble [54]. Therefore, evaluating the ζ-potential in the pH range studied is important. According to the results shown in Figure 1, ζ-potential values shifted from positive to negative values from acid to basic pH, passing through zero at a pH value of 4.5. This value corresponds to the isoelectric point at which all charges are balanced and the solubility presents a minimum value.

The value of pH 2 was selected as the most suitable due to it presents the highest ζ-potential value, indicating that there will be more surface charges with more electrostatic interactions and, therefore, the emulsions will be more resistant to destabilization phenomena [55]. In addition to its suitable solubility to be used as emulsifier.

#### 3.2.2. Surface Tension

Figure 2A shows that surface tension values vary as a function of concentration at pH 2. This will allow selecting an appropriate protein isolate concentration to prepare the emulsions. Figure 2B shows the variation in surface tension values as a function of pH values at a constant protein concentration value of 2.0 wt%.

Surface tension presents certain instability at low concentrations, decreasing the surface tension value as protein concentration increases, which implies an increase in emulsifying capacity by increasing the ability to reach the interface. This surface tension presents a constant value from 2 wt%, as the variation is not significant. In this sense, a protein concentration of 2 wt% seems to be the best option to process the emulsion. On the other hand, surface tension seems to increase while going from acidic to basic pH, thus it will be less stable at basic pH than at acidic pH. Therefore, as surface tension reaches the lowest value at pH 2, together with the ζ-potential test, this value seems to be the best option to manufacture the nanoemulsion. Similar results have been achieved in previous studies carried out by other authors [56,57,58].

#### 3.2.3. Interfacial Rheology

Figure 3 shows the values of the interfacial elastic and viscous moduli (G′ and G_i_″, respectively) as a function of frequency for the different systems tested (0.5 wt%, 1 wt% and 2 wt% isolate protein concentration) at a constant pH value of 2.

It can be observed that the viscoelastic moduli of the three systems are very stable to frequency variations, with a predominant elastic behavior in all three cases, which is a very typical behavior in protein-based systems [47]. It can also be seen that the systems with a higher protein concentration have higher values of viscoelastic moduli as the protein acts as an emulsifier. This is a positive aspect, since, as the interface is structured, the emulsions become more stable due to the formation of stronger intermolecular interactions [59].

For a better comparison, Table 3 shows the interfacial elastic moduli and loss tangents at 0.1 Hz (G′_i_^1^ and tanδ_i_^1^, respectively) for all studied systems. Thus, it is observed that higher protein contents generate an increment in elastic moduli, being more significant from 0.5 to 1.0 wt%. In addition, the values of the loss tangent for all the systems are below 1 and near 0, thereby showing that these systems have a solid-like behavior, which is also more relevant as the concentration increases [60].

### 3.3. Characterization of the Nanoemulsions

#### 3.3.1. Backscattering

The results of the backscattering at different pressures (100 PSI, 120 PSI and 140 PSI) for the system made with 2.0 wt% of soy protein isolate and pH 2 (selected conditions in the previous characterization) are shown in Figure 4A, Figure 4B and Figure 4C, respectively. Figure 4D shows the evolution of the relative BS as a function of time obtained within the middle part of the tube defined as follows:BS_rel_ = BS(t)/BS_0_(2)
where BS_0_ and BS(t) are the mean values for the BS profile obtained at the initial time and time *t*, respectively.

As can be observed, the nanoemulsion obtained with a pressure of 100 PSI (Figure 4A) remains stable until approximately 168 h after its processing, when the appearance of coalescence and creaming effect are observed. A pressure of 140 PSI (Figure 4C) generates an outlet temperature that causes an induced coalescence effect, which, in turn, reduces the stability of the nanoemulsion and, in addition, the appearance of certain amount of sedimentation is also observed. Regarding an intermediate pressure (Figure 4B), 120 PSI, the graph shows that there is no effect that destabilizes the nanoemulsion and, therefore, it seems to be the most adequate pressure to obtain it. In this way, relative BS (Figure 4D) remained constant when a pressure of 120 PSI was selected, while it decreased in the other two pressures, due to a coalescence effect. These results could be due to the fact that 100 PSI is an insufficient pressure to achieve an emulsion with a distribution of small and homogeneous droplets (between 20 and 200 nm), causing its stability to decrease and the droplets to coalesce easily. On the other hand, the pressure of 140 PSI is too high (related to high energy and, consequently, high temperatures), as it does not allow for the creation of a stable interface, causing the emulsion to also present instability [61].

#### 3.3.2. Droplet Size Distribution

The evolution of the emulsion droplet distribution (DSD) for the system processed at 120 PSI is shown in Figure 5.

As can be observed, there are no significant differences in the DSD over time, showing an almost normal unimodal distribution. However, over time, the emulsion becomes more unstable, increasing the dispersity and the droplet size, as can also be observed in Table 4. The uniformity of the distribution can be seen using parameter U (Table 4), since, the lower the U, the higher the uniformity of the system. Comparing the values over time, up to 7 days, there are no significant differences, whereas, at 19 days, the U value increased significantly. Nevertheless, Sauter’s diameter (D_3,2_) remains constant in the whole period of evaluation without significant differences between days. However, the volumetric diameter (D_4,3_) increases significantly, which indicates that growth in droplet size is occurring at high values, thus increasing the dispersity as a consequence of coalescence.

#### 3.3.3. Rheology

Figure 6 shows the frequency sweep for the system with 2.0 wt% and pH 2 over time (1 day, 7 days and 15 days).

As can be observed, the values of the elastic modulus are much higher than those of the viscous modulus, indicating that the nanoemulsion exhibits a gel-like behavior, despite being a beverage (Appendix A). This behavior is typical in emulsions where soy protein is used as emulsifier due to the bridging effect of the soy protein isolate. Moreover, the graph shows that the emulsion seems to be stable over the frequency for 15 days, with a slight decrease at low frequencies, related to a certain destabilization over a long period of time [62].

## 4. Conclusions

A method to prepare O/W nanoemulsion to encapsulate bioactive compounds using only soy protein isolate as emulsifier was developed in this study, showing good stability over time. These emulsions exhibited a gel-like behavior, characterized by G′ being over G″ within the experimental frequency range. Moreover, it was demonstrated that the 1:9 oil-water ratio provides good stability over time, enabling the development of a beverage (nanoemulsion) suitable for human consumption.

Knowledge of the properties of the aqueous phase with the emulsifier is a key factor for the development of nanoemulsions. A value of 2 wt% and pH 2 in the aqueous phase leads to lower values of surface tension and appropriate values of surface charges, which guarantee the success of emulsion manufacture. In addition, working at these values of pH provides microbiological stability. Regarding the processing pressure, a value of 120 PSI is the most suitable pressure, as no destabilization mechanism was observed. Lower values tend to form emulsions with larger droplet sizes, meaning that emulsions are less stable, and higher values could have granted greater stability. However, the outlet temperature produces an induced coalescence effect that destabilizes the emulsion.

Nevertheless, this work is a preliminary study and, although stable nanoemulsions were achieved, which may have a potential application for the incorporation of functional ingredients, a more in-depth evaluation of the incorporation of these ingredients and their behavior is necessary.

## Figures and Tables

**Figure 1 foods-12-00507-f001:**
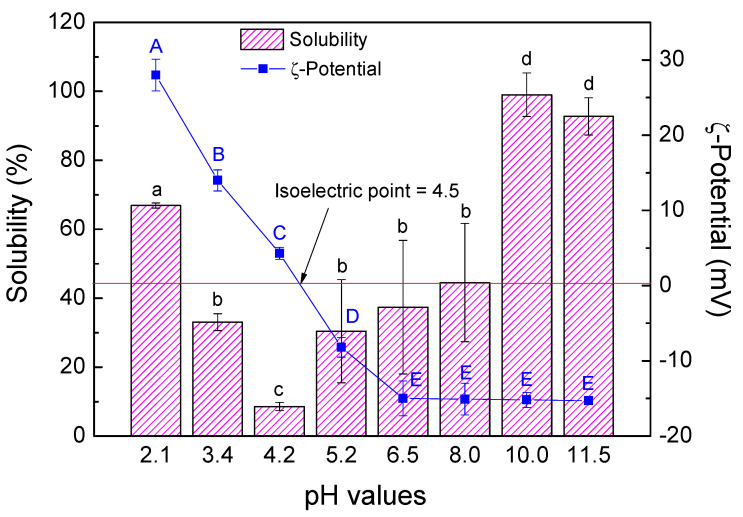
Solubility and ζ-potential values as a function of pH values. Different letters show significant differences in the values for solubility (a–d) and the values for ζ-Potential (A–E) (*p <* 0.05).

**Figure 2 foods-12-00507-f002:**
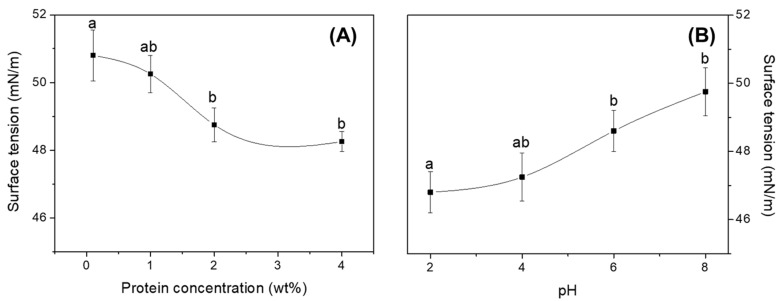
Surface tension values as a function of protein concentration at pH 2 (**A**) and as a function of pH at a protein isolate concentration of 2 wt% (**B**). Different letters show significant differences (*p <* 0.05).

**Figure 3 foods-12-00507-f003:**
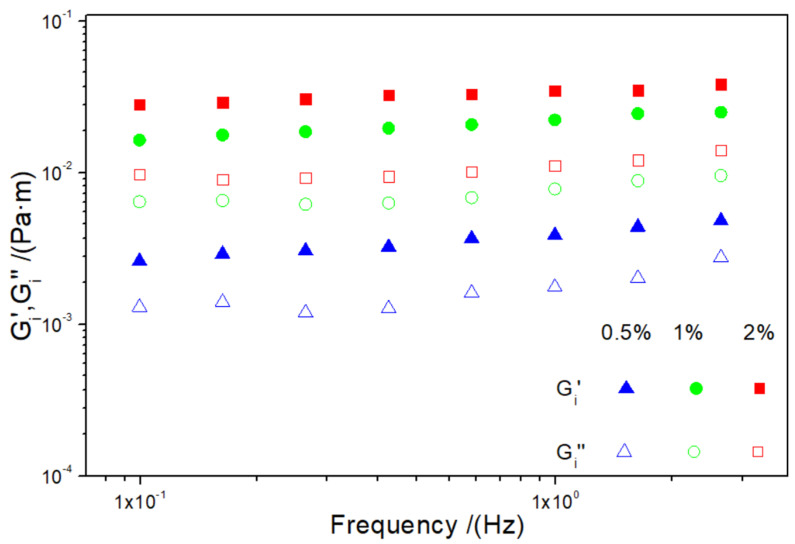
Elastic (G′_i_) and viscous (G″_i_) moduli as function of frequency for the three systems tested at pH 2.

**Figure 4 foods-12-00507-f004:**
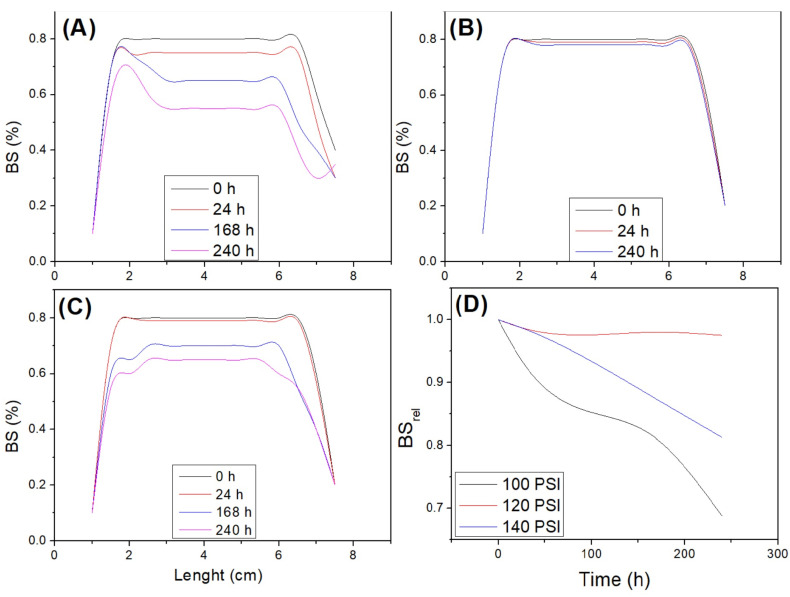
Backscattering (BS) measurements for the system processed at 100 PSI (**A**), 120 PSI (**B**) and 140 PSI (**C**). Relative backscattering (BS_rel_) for the three systems obtained was also included (**D**).

**Figure 5 foods-12-00507-f005:**
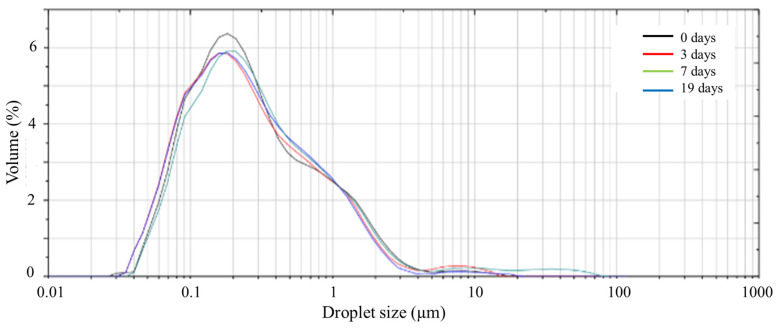
Evolution of droplet size distribution at different days of the emulsion at 2.0 wt% of soy protein isolate, pH 2 and processed at 120 PSI.

**Figure 6 foods-12-00507-f006:**
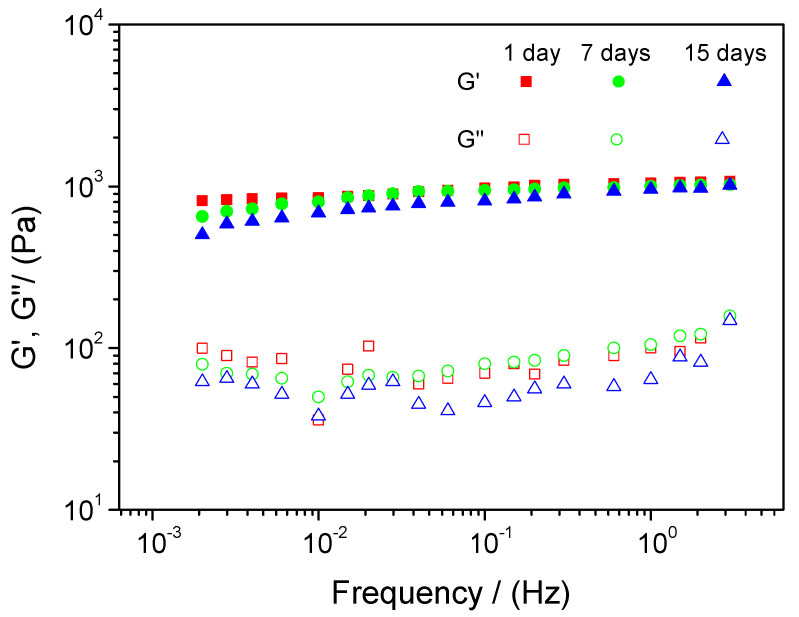
Frequency sweep for an emulsion at 2.0 wt% of soy protein isolate, pH 2 and processed at 120 PSI.

**Table 1 foods-12-00507-t001:** Chemical composition of the soy protein isolate. Different letters mean significant differences (*p* < 0.05).

	wt% Dry Basis
Moisture	6.0 ± 2.3 ^a^
Lipids	1.0 ± 0.5 ^b^
Carbohydrates	0.1 ± 0.3 ^b^
Ashes	5.0 ± 1.2 ^a^
Proteins	90.0 ± 1.0 ^c^

**Table 2 foods-12-00507-t002:** Amino acid composition of the soy protein isolate. * Hydrophilic groups. ** Hydrophobic groups.

Amino Acids	Concentration (%)
* Aspartic acid(Asp)	8.2
* Threonine (Thr)	2.5
* Serine (Ser)	1.0
* Glutamic acid (Glu)	21.7
* Glycine (Gly)	7.4
** Alanine (Ala)	10.4
* Cysteine (Cys)	1.7
** Valine (Val)	5.9
Methionine (Met)	0.9
** Isoleucine (Ile)	4.4
** Leucine (Leu)	8.6
Tryptophan (Trp)	<0.1
* Tyrosine (Tyr)	3.0
Phenylalanine (Phe)	4.2
* Histidine (His)	1.4
* Lysine (Lys)	5.6
Proline (Pro)	7.8
* Arginine (Arg)	5.3

**Table 3 foods-12-00507-t003:** Interfacial elastic modulus (G′_i_) and interfacial loss tangent (tan δ_i_) at 0.1 Hz for the three systems tested. Different letters show significant differences (*p <* 0.05).

	0.5 wt%	1.0 wt%	2.0 wt%
G′_i_^1^ (mPa·m)	3.71 ^a^	20.89 ^b^	33.10 ^c^
tan δ_i_^1^ (-)	0.437 ^B^	0.330 ^A^	0.309 ^A^

**Table 4 foods-12-00507-t004:** Mean values of particle diameters and uniformity for an emulsion at 2.0 wt% of soy protein isolate, pH 2 and processed at 120 PSI. Different letters show significant differences (*p <* 0.05).

t (Days)	D_3,2_ (nm)	D_4,3_ (nm)	U
0	162 ± 2.1 ^a^	459 ± 7 ^A^	1.6 ^α^
3	174 ± 1.4 ^b^	550 ± 6 ^C^	1.8 ^α^
7	173 ± 10 ^ab^	490 ± 4 ^B^	1.6 ^α^
19	171 ± 14 ^ab^	922 ± 122 ^D^	3.6 ^β^

## Data Availability

Not applicable.

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
