# Peer review of "Soy Protein Isolate as Emulsifier of Nanoemulsified Beverages: Rheological and Physical Evaluation"

_foods, 2023, doi:10.3390/foods12030507_

Round 1

Reviewer 1 Report

In this manuscript, the authors studied the design and evaluation of nanoemulsions. The characterization is not well correlated with the stability. 

1.      The meaning of a-d and A-E in Fig. 1 should be specified.

2.      As the surface tension varies a little, there is no reason to regard 2 wt% as the best option to process the emulsion.

3.      Does pH affect the interfacial rheology?

4.      How to obtain the parameter U in table 4?

5.      The oil content is around 10%. It is not usual to see a solid-like behavior (Fig. 6) in oil-in-water emulsion at this concentration. A possible mechanism is the bridging effect of the soy protein isolate.

6.      There is no direct correlation between the interfacial rheology and the stability of emulsion because the interfacial rheology is of solution-air interface instead of the water-oil interface. Moreover, the surface tension is also not helpful for the same reason. The interfacial tension between oil-water in presence of soy protein isolate should be supplied.

7.      There is no data for the effect of pH on the emulsion stability.

Reviewer 2 Report

Title:

It’s not suitable. The title of manuscript is general. Please specified your innovation in the title. Which bioactive components?

Abstract:

Rewrite this section carefully.

What’s your aims?

Which methods were used?

What’s your results?

Material and methods:

The references of methods is missing.

The zeta potential of emulsion should be added.

The physical stability should be added.

Particle size should be evaluated during the storage.

Result and discussion

Discussion should be improved based on new articles.
